# EvolMPNN: Predicting Mutational Effect on Homologous Proteins by Evolution Encoding

## Abstract

Predicting protein properties is paramount for biological and medical advancements. Current protein engineering mutates on a typical protein, called the *wild-type*, to construct a family of homologous proteins and study their properties. Yet, existing methods easily neglect subtle mutations, failing to capture the effect on the protein properties. To this end, we propose EvolMPNN, Evolution-aware Message Passing Neural Network, to learn evolution-aware protein embeddings. EvolMPNN samples sets of anchor proteins, computes evolutionary information by means of residues and employs a differentiable evolution-aware aggregation scheme over these sampled anchors. This way EvolMPNN can capture the mutation effect on proteins with respect to the anchor proteins. Afterwards, the aggregated evolution-aware embeddings are integrated with sequence embeddings to generate final comprehensive protein embeddings. Our model shows up to $6.4\%$ better than state-of-the-art methods and attains $36\times$ inference speedup in comparison with large pre-trained models. The code and models are available at https://anonymous.4open.science/r/EvolMPNN.

## 1 Introduction

*Can we predict important properties of a protein by directly observing only the effect of a few mutations on such properties?* This basic biological question (Wells, 1990; Fowler & Fields, 2014) has recently engaged the machine learning community due to the current availability of benchmark data (Rao et al., 2019; Dallago et al., 2021; Xu et al., 2022). Proteins are sequences of amino-acids (residues), which are the cornerstone of life and influence a number of metabolic processes, including diseases (Pauling et al., 1951; Ideker & Sharan, 2008). For this reason, protein engineering stands at the forefront of modern biotechnology, offering a remarkable toolkit to manipulate and optimise existing proteins for a wide range of applications, from drug development to personalised therapy (Ulmer, 1983; Carter, 2011; Alley et al., 2019).

One fundamental process in protein engineering progressively mutates an initial protein, called the *wild-type*, to study the effect on the protein's properties (Siezen et al., 1991). These mutations form a family of *homologous proteins* as in Figure 1. This process is appealing due to its cheaper cost compared to other methods and reduced time and risk (Wang et al., 2012; Engqvist & Rabe, 2019).

Yet, the way mutations affect the protein's properties is not completely understood (Bryant et al., 2021; Sarkisyan et al., 2016; Wu et al., 2016), as it depends on a number of chemical reactions and bonds among residues. For this reason, machine learning offers a viable alternative to model complex interactions among residues. Initial approaches employed *feature engineering* to capture protein's evolution (Saravanan & Gautham, 2015; Feng & Zhang, 2000); yet, a manual approach is expensive and does not offer enough versatility. Advances in NLP and CV inspired the design of deep *protein sequence encoders* (Hochreiter & Schmidhuber, 1997; Yu et al., 2017; Vaswani et al., 2017) and general purpose Protein Language Models (PLMs) that are pre-trained on large scale datasets of sequences. Notable PLMs include ProtBert (Brandes et al., 2022), AlphaFold (Jumper et al., 2021), TAPE Transformer (Rao et al., 2019) and ESM (Rives et al., 2021). These models mainly rely on Multiple Sequence Alignments (MSAs) (Meier et al., 2021) to search on large databases of protein evolution. Nevertheless, this search process is insensitive to subtle yet crucial mutations and introduces additional computational burdens (Pearson, 2013; Chatzou et al., 2016).

To overcome the limitations of previous models, we propose EVOLMPNN, Evolution-aware Message Passing Neural Network, to predict the mutational effect on homologous proteins. Our fundamental assumption is that there are inherent correlations between protein properties and the sequence differences among them, as shown in Figure 1-(b). EVOLMPNN integrates both protein sequence and evolutionary information by identifying where and which mutations occur on the target protein sequence, compared with known protein sequences and predicts the mutational effect on the target protein property. To avoid the costly *quadratic* pairwise comparison among proteins, we devise a theoretically grounded (see Section 4.6) *linear* sampling strategy to compute differences only among the proteins and a fixed number of anchor proteins (Section 4.2). We additionally introduce two extensions of our model, EVOLGNN and EVOLFORMER, to include available data on the relation among proteins (Section 4.5). The theoretical computation complexity of proposed methods are provided to guarantee their efficiency and practicality. We apply the proposed methods to three benchmark homologous protein property prediction datasets with nine splits. Empirical evaluation results (Section 5.1) show up to $6.7\%$ Spearman's $\rho$ correlation improvement over the best performing baseline models, reducing the inference time by $36\times$ compared with pre-trained PLMs.

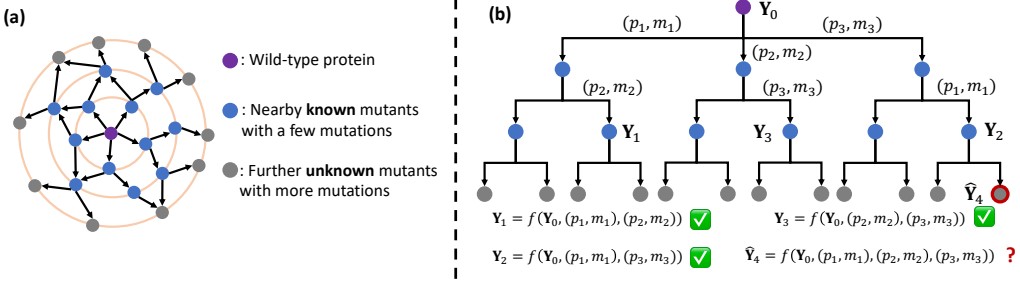

Figure 1: Protein property prediction on homologous protein family. (a) An example homologous protein family with labelled nearby mutants with few mutations. The task is to predict the label of unknown mutants with more mutations. (b) The evolutionary pattern for (a); $(p_1, m_1)$ indicates the mutation $m_1$ at the $p_1$-th position of the protein sequence.

## 2   PRELIMINARY AND PROBLEM

In the protein engineering process, we first receive a *set of proteins* $\mathcal{M} = \{\mathcal{P}_i\}_{i=1,2,...,M}$ in which each protein can be associated with a label vector $\mathbf{Y}_i \in \mathbb{R}^\theta$ that describes its biomedical properties, *e.g.*, fitness, stability, fluorescence, solubility, etc. Each protein $\mathcal{P}_i$ is a linear chain of *amino-acids* $\mathcal{P}_i = \{r_j\}_{j=1,2,...,N}$. While a protein sequence folds into specific 3D conformation to perform some biomedical functions, each amino-acid is considered as a *residue*. Residues are connected to one another by peptide bonds and can interact with each other by different chemical bounds (Pauling et al., 1951). In short, the function of a protein is mainly determined by the chemical interactions between residues. Since the 3D structure is missing in benchmark datasets (Rao et al., 2019; Dallago et al., 2021; Xu et al., 2022), we assume no 3D protein information in this paper.

**Homologous Protein Family.** A set of protein sequences ($\mathcal{M}$) is a *homologous protein family* if there exists an ancestral protein $\mathcal{P}_{\text{WT}}$, called *wild-type*, such that any $\mathcal{P}_i \in \mathcal{M}$ is obtained by mutating $\mathcal{P}_{\text{WT}}$ through substitution, deletion, insertion and truncation of residues (Ochoterena et al., 2019). As shown in Figure 1-(a), a homologous protein family can be organised together by representing their evolutionary relationships and Figure 1-(b) illustrates the detailed evolutionary patterns.

**Research Problem.** Protein engineering based on homologous proteins is a promising and essential direction for designing novel proteins of desired properties (Huang et al., 2014; Langan et al., 2019). Understanding the relation between protein sequence and property is one essential step. Practically, biologists perform experiments in the lab to label the property $\mathbf{Y}_{\text{TRAIN}}$ of a set of protein $\mathcal{M}_{\text{TRAIN}} \subset \mathcal{M}$ and the follow-up task is predicting $\hat{\mathbf{Y}}_{\text{TEST}}$ of the rest proteins $\mathcal{M}_{\text{TEST}} \subset \mathcal{M}$. However, homologous proteins typically have similarities in their amino-acid sequences, structures, and functions due to their shared ancestry. Accurately predicting the homologous protein property by distinguishing these subtle yet crucial differences is still an open challenge.

## 3 RELATED WORK

**Feature Engineering.** Besides conducting manual experiments in labs to measure protein properties, the basic solution is to design different feature engineering methods based on relevant biological knowledge, to extract useful information from protein sequence (Klein et al., 1985; Feng & Zhang, 2000; Wang et al., 2017). Dallago et al. (2021) introduce using Levenshtein distance (Li & Liu, 2007) and BLOSUM62-score (Eddy, 2004) relative to wild-type to design protein sequence features. In another benchmark work, Xu et al. (2022) adopt another two typical protein sequence feature descriptors, *i.e.*, Dipeptide Deviation from Expected Mean (DDE) (Saravanan & Gautham, 2015) and Moran correlation (Moran) (Feng & Zhang, 2000). For more engineering methods, refer to the comprehensive review (Lee et al., 2007).

**Protein Representation Learning.** In the last decades, empowered by the outstanding achievements of machine learning and deep learning, protein representation learning has revolutionised protein property prediction research. Early work along this line adopts the idea of word2vec (Mikolov et al., 2013) to protein sequences (Xu et al., 2018; Mejía-Guerra & Buckler, 2019). To increase model capacity, deeper *protein sequence encoders* were proposed by the Computer Vision (CV) and Nature Language Processing (NLP) communities (Hochreiter & Schmidhuber, 1997; Yu et al., 2017; Vaswani et al., 2017). The latest works develop *Protein Language Models* (PLMs), which focus on employing deep sequence encoder models for protein sequences and are pre-trained on million- or billion-scale sequences. Well-known works include ProtBert (Brandes et al., 2022), AlphaFold (Jumper et al., 2021), TAPE Transformer (Rao et al., 2019) and ESM (Rives et al., 2021). However, most existing work does not pay enough attention to these subtle yet crucial differences in homologous proteins. Rives et al. (2021); Jumper et al. (2021) explore protein Multiple Sequence Alignments (MSAs) (Rao et al., 2021a; Meier et al., 2021) to capture the mutational effect. Nevertheless, the MSA searching process introduces additional computational burden and is insensitive to subtle but crucial sequence differences (Pearson, 2013). Chatzou et al. (2016) indicate the shortcomings of MSAs on easily neglecting the presence of minor mutations, which can propagate errors to downstream protein sequence representation learning tasks. This paper also lies in this direction, we propose a novel solution for the challenging homologous protein property prediction tasks.

## 4 FRAMEWORK

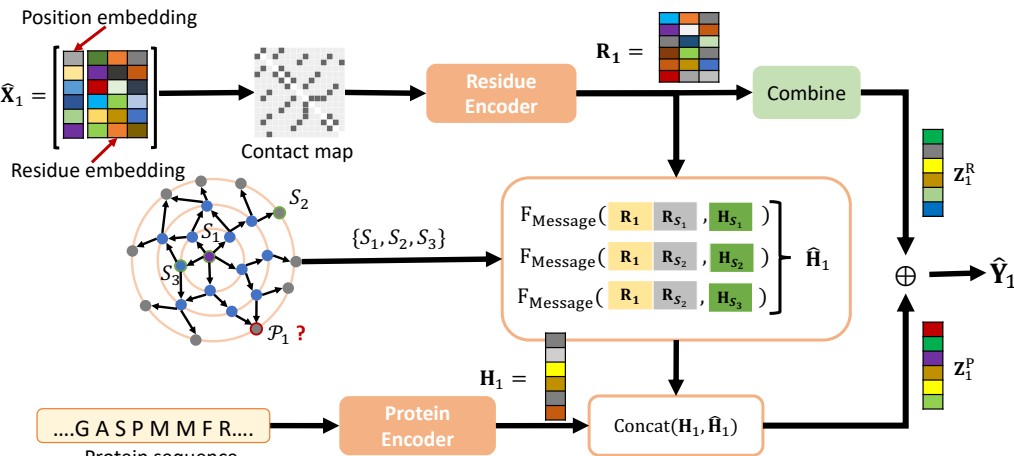

Figure 2: Our EVOLMPNN framework encodes protein mutations via a sapient combination of residue evolution and sequence encoding.

EVOLMPNN is a novel framework that integrates both protein sequence information and evolution information by means of residues. As a result, EVOLMPNN accurately predicts the mutational effect on homologous protein families. First, in Section 4.1, we introduce *embedding initialisation* for protein sequence and residues and the update module for residue embedding (Section 4.2). The *evolution encoding* in Section 4.3 is the cornerstone of the model that ameliorates protein embeddings.

We conclude in Section 4.4 with the generation of *final proteins embeddings and model optimisation*. We complement our model with a theoretical analysis to motivate our methodology and a discussion of the computation complexity (Section 4.6). We additionally propose extended versions of EVOLMPNN that deal with available protein-protein interactions (Section 4.5).

## 4.1 EMBEDDING INITIALISATION

**Protein Sequence Embedding.** Given a *set of proteins* $\mathcal{M} = \{\mathcal{P}_i\}_{i=1,2,...,M}$, we first adopt a (parameter-frozen) PLM model (Rao et al., 2021b; Meier et al., 2021)[1] as protein sequence encoder to initialise the protein sequence embedding ($\mathbf{H}$) for every protein $\mathcal{P}_i$, which include *macro* (*i.e.*, protein sequence) level information as the primary embedding.

$$\mathbf{H} = \text{PLMENCODER}(\{\mathcal{P}_i\}_{i=1,2,...,M}),\tag{1}$$

where the obtained protein embedding $\mathbf{H} \in \mathbb{R}^{M \times d}$ and $\mathbf{H}_i$ corresponds to each protein $\mathcal{P}_i$. Different encoders can extract information on various aspects, however, existing PLM models that rely on MSAs are not sensitive enough to capture the evolution pattern information in homologous protein families (Pearson, 2013). Chatzou et al. (2016) systematically indicate the shortcomings of MSAs on easily neglecting the presence of minor mutations, which can propagate errors to downstream protein sequence representation learning tasks.

**Residue Embedding.** In order to properly capture the evolution information in homologous proteins, we delve into the residue level for *micro* clues. We adopt two residue embedding initialisation approaches, *i.e.*, one-hot encoding ($\Phi^{\text{OH}}$) and pre-trained PLM encoder ($\Phi^{\text{PLM}}$), to generate protein's initial residue embeddings $\mathbf{X}_i = \{\mathbf{x}_j^i\}_{j=1,2,...,N}$, where $\mathbf{x}_j^i \in \mathbb{R}^d$. In particular, $\Phi^{\text{OH}}$ assigns each protein residue[2] with a binary feature vector $\mathbf{x}_j^i$, where $\mathbf{x}_{jb}^i = 1$ indicates the appearance of the $b$-th residue at $\mathcal{P}_i$'s $j$-th position. By stacking $N$ residues' feature vectors into a matrix, we can obtain $\mathbf{X}_i \in \mathbb{R}^{N \times d}$. On the other hand, following the benchmark implementations (Zhu et al., 2022), PLMENCODER can export residue embeddings similar to Eq. 1. Formally, $\Phi^{\text{PLM}}$ initialises protein residue embeddings as $\mathbf{X}_i = \text{PLMENCODER}(\{r_j\}_{j=1,2,...,N})$.

**Position Embedding.** Another essential component of existing PLM is the positional encoding, which was first proposed by Vaswani et al. (2017). This positional encoding effectively captures the relative structural information between entities and integrates it with the model (Ying et al., 2021). In our case, correctly recording the position of each residue in the protein sequence plays an essential role in identifying each protein's corresponding mutations. Because the same mutation that occurs at different positions may lead to disparate influences on protein property. Therefore, after initialising residue embeddings, we further apply positional embedding on each protein's residues. We adopt a methodology that reminisces (Ruoss et al., 2023) that demonstrates the paramount importance of assigning each residue with a unique position embedding. As such, we randomly initialise a set of $d$ position embeddings $\Phi^{\text{Pos}} \in \mathbb{R}^{N \times d}$. We denote the residue embedding empowered by position embedding as $\hat{\mathbf{X}}_i = \mathbf{X}_i \odot \Phi^{\text{Pos}}$.

## 4.2 RESIDUE EMBEDDING UPDATE

The 3D protein folding depends on the strength of different chemical bonds between residues to maintain a stable 3D structure. Previous studies carefully designed residue contact maps to model the residue-residue interactions to learn effective residue embeddings (Rao et al., 2021b; Gao et al., 2023). In this paper, we adopt the residue-residue interaction to update residue embeddings but eschew the requirement of manually designing the contact map. Instead, we assume the existence of an implicit fully connected residue contact map of each protein $\mathcal{P}_i$ and implement the Transformer model (Vaswani et al., 2017; Wu et al., 2022) to adaptively update residue embeddings. Denote $\mathbf{R}_i^{(\ell)}$ as the input to the $(\ell + 1)$-th layer, with the first $\mathbf{R}_i^{(0)} = \hat{\mathbf{X}}_i$ be the input encoding. The $(\ell + 1)$-th

---

[1]We do not fine-tune PLM in this paper for efficiency consideration.
[2]There are 20 different amino-acid residues commonly found in proteins

layer of residue embedding update module can be formally defined as follows:

$$\mathbf{Att}_i^h(\mathbf{R}_i^{(\ell)}) = \text{SOFTMAX}(\frac{\mathbf{R}_i^{(\ell)}\mathbf{W}_Q^{\ell,h}(\mathbf{R}_i^{(\ell)}\mathbf{W}_K^{\ell,h})^{\mathrm{T}}}{\sqrt{d}}),$$

$$\hat{\mathbf{R}}_i^{(\ell)} = \mathbf{R}_i^{(\ell)} + \sum_{h=1}^{H}\mathbf{Att}_i^h(\mathbf{R}_i^{(\ell)})\mathbf{R}_i^{(\ell)}\mathbf{W}_V^{\ell,h}\mathbf{W}_O^{\ell,h}, \tag{2}$$

$$\mathbf{R}_i^{(\ell+1)} = \hat{\mathbf{R}}_i^{(\ell)} + \text{ELU}(\hat{\mathbf{R}}_i^{(\ell)}\mathbf{W}_1^\ell)\mathbf{W}_2^\ell,$$

where $\mathbf{W}_O^{\ell,h} \in \mathbb{R}^{d_H \times d}$, $\mathbf{W}_Q^{l,h}$, $\mathbf{W}_K^{l,h}$, $\mathbf{W}_V^{l,h} \in \mathbb{R}^{d \times d_H}$, $\mathbf{W}_1^\ell \in \mathbb{R}^{d \times r}$, $\mathbf{W}_2^\ell \in \mathbb{R}^{d_t \times d}$, $H$ is the number of attention heads, $d_H$ is the dimension of each head, $d_t$ is the dimension of the hidden layer, ELU (Clevert et al., 2015) is an activation function, and $\mathbf{Att}_i^h(\mathbf{R}_i^{(\ell)})$ refers to as the attention matrix. After each Transformer layer, we add a normalisation layer *i.e.*, LayerNorm (Ba et al., 2016), to reduce the over-fitting problem proposed by Vaswani et al. (2017). After stacking $L_r$ layers, we obtain the final residue embeddings as $\mathbf{R}_i = \mathbf{R}_i^{(L_r)}$.

## 4.3 EVOLUTION ENCODING

In homologous protein families, all proteins are mutants derived from a common wild-type protein $\mathcal{P}_{\text{WT}}$ with different numbers and types of mutations. In this paper, we propose to capture the evolutionary information via the following assumption.

**Assumption 1** (Protein Property Relevance). *Assume there is a homologous protein family $\mathcal{M}$ and a function $\text{F}_{\text{DIFF}}$ can accurately distinguish the mutations on mutant $\mathcal{P}_i$ compared with any $\mathcal{P}_j$ as $\text{F}_{\text{DIFF}}(\mathcal{P}_i, \mathcal{P}_j)$. For any target protein $\mathcal{P}_i$, its property $\mathbf{Y}_i$ can be predicted by considering 1) its sequence information $\mathcal{P}_i$; 2) $\text{F}_{\text{DIFF}}(\mathcal{P}_i, \mathcal{P}_j)$ and the property of $\mathcal{P}_j$, i.e., $\mathbf{Y}_j$. Shortly, we assume there exists a function $f$ that maps $\mathbf{Y}_i \leftarrow f(\text{F}_{\text{DIFF}}(\mathcal{P}_i, \mathcal{P}_j), \mathbf{Y}_j)$.*

Motivated by Assumption 1, we take both protein sequence and the mutants difference $\text{F}_{\text{DIFF}}(\mathcal{P}_i, \mathcal{P}_j)$ to accurately predict the protein property. To encode the protein sequence, we employ established tools described in Section 4.1. Here instead, we describe the evolution encoding to realise the function of $\text{F}_{\text{DIFF}}(\mathcal{P}_i, \mathcal{P}_j)$.

The naïve solution to extract evolutionary patterns in a homologous family is constructing a complete phylogenetic tree (Fitch & Margoliash, 1967) based on the mutation distance between each protein pair. Yet, finding the most parsimonious phylogenetic tree is **NP**-hard (Sankoff, 1975).

To address the aforementioned problems, we propose an *anchor-based protein evolution encoding* method. Specifically, denote $\mathbf{H}_i^{(\ell)}$ as the input to the $(\ell+1)$-th block and define $\mathbf{H}_i^{(0)} = \mathbf{H}_i$. The evolution localisation encoding of the $(\ell+1)$-th layer contains the following key components: *(i)* $k$ anchor protein $\{\mathcal{P}_{S_i}\}_{i=1,2,...,k}$ selection. *(ii)* Evolutionary information encoding function $\text{F}_{\text{DIFF}}$ that computes the difference between residues of each protein and those of the anchor protein, and target protein's evolutionary information is generated by summarising the obtained differences:

$$\mathbf{d}_{ij} = \text{COMBINE}(\mathbf{R}_i - \mathbf{R}_{S_j}), \tag{3}$$

where COMBINE can be implemented as differentiable operators, such as, CONCATENATE, MAX POOL MEAN POOL and SUM POOL; here we use the MEAN POOL to obtain $\mathbf{d}_{ij} \in \mathbb{R}^d$. *(iii)* Message computation function $\text{F}_{\text{MESSAGE}}$ that combines protein sequence feature information of two proteins with their evolutionary differences. We empirically find that the simple element-wise product between $\mathbf{H}_j^{(\ell)}$ and $\mathbf{d}_{ij}$ attains good results

$$\text{F}_{\text{MESSAGE}}(i, j, \mathbf{H}_j^{(\ell)}, \mathbf{d}_{ij}) = \mathbf{H}_j^{(\ell)} \odot \mathbf{d}_{ij}, \tag{4}$$

*(iv)* Aggregate messages from $k$ anchors and combine them with protein's embedding as follow:

$$\hat{\mathbf{H}}_i^{(\ell)} = \text{COMBINE}(\{\text{F}_{\text{MESSAGE}}(i, j, \mathbf{H}_j^{(\ell)}, \mathbf{d}_{ij})\}_{j=1,2,...,k}), \tag{5}$$

$$\mathbf{H}_i^{(\ell+1)} = \text{CONCAT}(\mathbf{H}_i^{(\ell)}, \hat{\mathbf{H}}_i^{(\ell)})\mathbf{W}^\ell, \tag{6}$$

where $\mathbf{W}^\ell \in \mathbb{R}^{2d \times d}$ transform concatenated vectors to the hidden dimension. After stacking $L_p$ layers, we obtain the final protein sequence embedding $\mathbf{Z}_i^{\text{P}} = \mathbf{H}_i^{(L_p)}$.

## 4.4 FINAL EMBEDDING AND OPTIMISATION

After obtaining protein $\mathcal{P}_i$'s residue embeddings $\mathbf{R}_i$ and sequence embedding $\mathbf{Z}_i^{\mathrm{P}}$, we summarise its residue embeddings as a vector $\mathbf{Z}_i^{\mathrm{R}} = \mathrm{MEAN\ POOL}(\mathbf{R}_i)$. The final protein embedding summarises the protein sequence information and evolution information as the comprehensive embedding $\mathbf{Z}_i = \mathrm{CONCAT}(\mathbf{Z}_i^{\mathrm{P}}, \mathbf{Z}_i^{\mathrm{R}})$ and the final prediction is computed as $\hat{\mathbf{Y}}_i = \mathbf{Z}_i \mathbf{W}^{\mathrm{FINAL}}$ where $\mathbf{W}^{\mathrm{FINAL}} \in \mathbb{R}^{d \times \theta}$, $\theta$ is the number of properties to predict. Afterwards, we adopt a simple and common strategy, similar to (Xu et al., 2022), to solve the protein property prediction tasks. Specifically, we adopt the MSELoss ($\mathcal{L}$) to measure the correctness of model predictions on training samples against ground truth labels. The objective of learning the target task is to optimise model parameters to minimise the loss $\mathcal{L}$ on this task. The framework of EVOLMPNN is summarised in Algorithm 1 in Appendix A.

## 4.5 EXTENSIONS ON OBSERVED GRAPH

EVOLMPNN does not leverage any information from explicit geometry among proteins, where each protein only communicates with randomly sampled anchors (Section 4.3). However, it is often possible to have useful structured data $G = (\mathcal{M}, \mathbf{A})$ that represents the relation between protein-protein by incorporating specific domain knowledge (Zhong et al., 2023).[3] Therefore, here we introduce EVOLGNN, an extension of EVOLMPNN on the possibly observed protein interactions.

**EVOLGNN.** We compute the evolution information as Eq. 3. The evolution information can be easily integrated into the pipeline of message-passing neural networks, as an additional structural coefficient (Wijesinghe & Wang, 2022):

$$\mathbf{m}_a^{(\ell)} = \mathrm{AGGREGATE}^{\mathcal{N}}(\{\mathbf{A}_{ij}, \underbrace{\mathbf{d}_{ij}}_{\text{Evol. info.}}, \mathbf{H}_j^{(\ell-1)} \,|\, j \in \mathcal{N}(i)\}),$$

$$\mathbf{m}_i^{(\ell)} = \mathrm{AGGREGATE}^{\mathcal{I}}(\{\mathbf{A}_{ij}, \underbrace{\mathbf{d}_{ij}}_{\text{Evol. info.}} \,|\, j \in \mathcal{N}(i)\})\, \mathbf{H}_i^{(\ell-1)}, \quad (7)$$

$$\mathbf{H}_i^{(\ell)} = \mathrm{COMBINE}(\mathbf{m}_a^{(\ell)}, \mathbf{m}_i^{(\ell)}),$$

where $\mathrm{AGGREGATE}^{\mathcal{N}}(\cdot)$ and $\mathrm{AGGREGATE}^{\mathcal{I}}(\cdot)$ are two parameterised functions. $\mathbf{m}_a^{(\ell)}$ is a message aggregated from the neighbours $\mathcal{N}(i)$ of protein $\mathcal{P}_i$ and their structure ($\mathbf{A}_{ij}$) and evolution ($\mathbf{d}_{ij}$) coefficients. $\mathbf{m}_i^{(\ell)}$ is an updated message from protein $\mathcal{P}_i$ after performing an element-wise multiplication between $\mathrm{AGGREGATE}^{\mathcal{I}}(\cdot)$ and $\mathbf{H}_i^{(\ell-1)}$ to account for structural and evolution effects from its neighbours. After, $\mathbf{m}_a^{(\ell)}$ and $\mathbf{m}_i^{(\ell)}$ are combined together to obtain the update embedding $\mathbf{H}_i^{(\ell)}$.

**EVOLFORMER.** Another extension relies on pure Transformer structure, which means the evolution information of $\mathcal{M}$ can be captured by every protein. The evolution information can be integrated into the pipeline of Transformer, as additional information to compute the attention matrix:

$$\mathbf{Att}^h(\mathbf{H}^{(\ell)}) = \mathrm{SOFTMAX}\Big(\frac{\mathbf{H}^{(\ell)}\mathbf{W}_Q^{\ell,h}(\mathbf{H}^{(\ell)}\mathbf{W}_K^{\ell,h})^{\mathrm{T}}}{\sqrt{d}} + \underbrace{\mathrm{MEAN\ POOL}(\{\mathbf{R}_i\}_{i=1,2,...,M})}_{\text{Evol. info.}}\Big), \quad (8)$$

Other follow-up information aggregation and feature vector update operations are the same as the basic Transformer pipeline, as described in Eq. 2.

## 4.6 THEORETICAL ANALYSIS

**Anchor Selection.** Inspired by (You et al., 2019), we adopt Bourgain's Theorem (Bourgain, 1985) to guide the random anchor number ($k$) of the evolution encoding layer. Briefly, support by a constructive proof (Theorem 2 (Linial et al., 1995)) of Bourgain Theorem (Theorem 1), only $k = O(\log^2 M)$ anchors are needed to ensure the resulting embeddings are guaranteed to have low distortion (Definition 1), in a given metric space $(\mathcal{M}, \mathrm{F}_{\mathrm{DIST}})$. EVOLMPNN can be viewed as a generalisation of the embedding method of Theorem 2, where $\mathrm{F}_{\mathrm{DIST}}(\cdot)$ is generalised via message passing functions

---

[3]Available contact map describes residue-residue interactions can be easily integrated as relational bias of Transformer (Wu et al., 2022) as we used in Section 4.2.

(Eq 3-Eq. 6). Therefore, Theorem 2 offers a theoretical guide that $O(\log^2 M)$ anchors are needed to guarantee low distortion embedding. Following this principle, EVOLMPNN choose $k = \log^2 M$ random anchors, denoted as $\{S_j\}_{j=1,2,\ldots,\log^2 M}$, and we sample each protein in $\mathcal{M}$ independently with probability $\frac{1}{2^j}$. Detailed discussion and proof refer to Appendix B.

**Complexity Analysis.** The computation costs of EVOLMPNN, EVOLGNN, and EVOLFORMER come from residue encoding and evolution encoding, since the protein sequence and residue feature initialisation have no trainable parameters. The residue encoder introduces the complexity of $O(MN)$ following an efficient implementation of NodeFormer (Wu et al., 2022). In the evolution encoding, EVOLMPNN performs communication between each protein and $\log^2 M$ anchors, which introduces the complexity of $O(M \log^2 M)$; EVOLGNN performs communication between each protein and $K$ neighbours with $O(KM)$ complexity; EVOLFORMER performs communication between all protein pairs, which introduces the complexity of $O(M)$, following the efficient implement, NodeFormer. In the end, we obtain the total computation complexity of EVOLMPNN - $O((N + \log^2 M)M)$, EVOLGNN - $O((N + K)M)$ and EVOLFORMER - $O((N + 1)M)$.

## 5 EXPERIMENTAL STUDY

In this section, we empirically study the performance of EVOLMPNN. We validate our model on three benchmark homologous protein family datasets and evaluate the methods on nine data splits to consider comprehensive practical use cases. Our experiments comprise a comprehensive set of state-of-the-art methods from different categories. We additionally demonstrate the effectiveness of two extensions of our model, EVOLGNN and EVOLFORMER, with different input features. We conclude our analysis studying the influence of some hyper-parameters and investigating the performance of EVOLMPNN on high mutational mutants.

Table 1: Datasets splits, and corresponding statistics; if the split comes from a benchmark paper, we report the corresponding citation.

| Landscape | Split | # Total | #Train | #Valid | #Test |
|---|---|---|---|---|---|
| AAV (Bryant et al., 2021) | 2-VS-REST (Dallago et al., 2021) | 82,583 | 28,626 | 3,181 | 50,776 |
| | 7-VS-REST (Dallago et al., 2021) | 82,583 | 63,001 | 7,001 | 12,581 |
| | LOW-VS-HIGH (Dallago et al., 2021) | 82,583 | 42,791 | 4,755 | 35,037 |
| GB1 (Wu et al., 2016) | 2-VS-REST (Dallago et al., 2021) | 8,733 | 381 | 43 | 8,309 |
| | 3-VS-REST (Dallago et al., 2021) | 8,733 | 2,691 | 299 | 5,743 |
| | LOW-VS-HIGH (Dallago et al., 2021) | 8,733 | 4,580 | 509 | 3,644 |
| Fluorescence (Sarkisyan et al., 2016) | 2-VS-REST | 54,025 | 12,712 | 1,413 | 39,900 |
| | 3-VS-REST (Xu et al., 2022) | 54,025 | 21,446 | 5,362 | 27,217 |
| | LOW-VS-HIGH | 54,025 | 44,082 | 4,899 | 5,044 |

**Datasets and Splits.** We perform experiments on benchmark datasets of several important protein engineering tasks, including `AAV`, `GB1` and `Fluorescence`, and generate three splits on each dataset. Data statistics are summarised in Table 1. The split $\lambda$-VS-REST indicates that we train models on wild-type protein and mutants of no more than $\lambda$ mutations, while the rest are assigned to test. The split LOW-VS-HIGH indicates that we train models on sequences with target value scores equal to or below wild-type, while the rest are assigned to test. For more details refer to Appendix C.

**Baselines.** As baseline models, we consider methods in four categories. First, we selected four *feature engineer* methods, *i.e.*, Levenshtein (Dallago et al., 2021), BLOSUM62 (Dallago et al., 2021), DDE (Saravanan & Gautham, 2015) and Moran (Feng & Zhang, 2000). Second, we select four *protein sequence encoder* models, *i.e.*, LSTM (Hochreiter & Schmidhuber, 1997), Transformer (Rao et al., 2019), CNN (Rao et al., 2019) and ResNet (Yu et al., 2017). Third, we select four *pre-trained PLM models*, *i.e.*, ProtBert (Elnaggar et al., 2022), ESM-1b (Rives et al., 2021), ESM-1v (Meier et al., 2021) and ESM-2 (Lin et al., 2023). In the end, we select four *GNN-based methods* which can utilise available graph structure, *i.e.*, GCN (Kipf & Welling, 2017), GAT (Velickovic et al., 2018), GraphTransformer (Shi et al., 2021) and NodeFormer (Wu et al., 2022).

**Implementation.** We follow the PEER benchmark settings[4], including train and test pipeline, model optimisation and evaluation method (evaluation is Spearman's $\rho$ metric), adopted in (Xu et al., 2022) to make sure the comparison fairness. For the baselines, including feature engineer, protein sequence

---

[4]https://github.com/DeepGraphLearning/PEER_Benchmark

encoder and pre-trained PLM, we adopt the implementation provided by benchmark Torchdrug (Zhu et al., 2022) and the configurations reported in (Xu et al., 2022). For the GNN-based baselines, which require predefined graph structure and protein features, we construct $K$-NN graphs (Eppstein et al., 1997), with $K = \{5, 10, 15\}$, and report the best performance. As features, we use the trained sequence encoder, which achieves better performance, used also in our method. In addition, we adopt ESM-1b as the residue encoder on `GB1` dataset and adopt One-Hot encoding on `AAV` and `Fluorescence` datasets to speed up the training process. All experiments are conducted on two NVIDIA GeForce RTX 3090 GPUs with 24 GB memory, and we report the mean performance of three runs with different random seeds. We present more details in Appendix D. Note that we do not report results that take more than 48 hours due to our limited computation resources.

## 5.1 EFFECTIVENESS

Table 2: Quality in terms Spearman's $\rho$ correlation with target value. NA indicates a non-applicable setting. * Used as a feature extractor with pre-trained weights frozen. † Results reported in (Dallago et al., 2021; Xu et al., 2022). - Can not complete the training process within 48 hours on our devices. Top-2 performances of each split are marked as **bold** and underline.

| Category | Model | AAV | | | GB1 | | | Fluorescence | | |
|---|---|---|---|---|---|---|---|---|---|---|
| | | 2-vs-R. | 7-vs-R. | L.-vs-H. | 2-vs-R. | 3-vs-R. | L.-vs-H. | 2-vs-R. | 7-vs-R. | L.-vs-H. |
| **Feature Engineer** | Levenshtein | 0.578 | 0.550 | 0.251 | 0.156 | -0.069 | -0.108 | 0.466 | 0.054 | 0.011 |
| | BLOSUM62 | NA | NA | NA | 0.128 | 0.005 | -0.127 | NA | NA | NA |
| | DDE | 0.649† | 0.636 | 0.158 | 0.445† | 0.816 | 0.306 | 0.690 | 0.638† | 0.159 |
| | Moran | 0.437† | 0.398 | 0.069 | 0.069† | 0.589 | 0.193 | 0.445 | 0.400† | 0.046 |
| **Protein Seq. Encoder** | LSTM | 0.125† | 0.608 | 0.308 | -0.002† | -0.002 | -0.007 | 0.256 | 0.494† | 0.207 |
| | Transformer | 0.681† | 0.748 | 0.304 | 0.271† | 0.877 | 0.474 | 0.250 | 0.643† | 0.161 |
| | CNN | 0.746† | 0.730 | 0.406 | 0.502† | 0.857 | 0.515 | 0.805 | 0.682† | **0.249** |
| | ResNet | 0.739† | 0.733 | 0.223 | 0.133† | 0.542 | 0.396 | 0.594 | 0.636† | 0.243 |
| **Pre-trained PLM** | ProtBert | 0.794† | - | - | 0.634† | 0.866 | 0.308 | 0.451 | 0.679† | - |
| | ProtBert* | 0.209† | 0.507 | 0.277 | 0.123† | 0.619 | 0.164 | 0.403 | 0.339† | 0.161 |
| | ESM-1b | 0.821† | - | - | 0.704† | 0.878 | 0.386 | 0.804 | 0.679† | - |
| | ESM-1b* | 0.454† | 0.573 | 0.241 | 0.337† | 0.605 | 0.178 | 0.528 | 0.430† | 0.091 |
| | ESM-1v* | 0.533 | 0.580 | 0.171 | 0.359 | 0.632 | 0.180 | 0.562 | 0.563 | 0.070 |
| | ESM-2* | 0.475 | 0.581 | 0.199 | 0.422 | 0.632 | 0.189 | 0.501 | 0.511 | 0.084 |
| **GNN-based Methods** | GCN | 0.824 | 0.730 | 0.361 | 0.745 | 0.865 | 0.466 | 0.755 | 0.677 | 0.198 |
| | GAT | 0.821 | 0.741 | 0.369 | 0.757 | 0.873 | 0.508 | 0.768 | 0.667 | 0.208 |
| | GraphTransf. | 0.827 | 0.749 | 0.389 | 0.753 | 0.876 | 0.548 | 0.780 | 0.678 | 0.231 |
| | NodeFormer | 0.827 | 0.741 | 0.393 | 0.757 | 0.877 | 0.543 | 0.794 | 0.677 | 0.213 |
| **Ours** | EVOLMPNN | **0.835** | **0.757** | **0.433** | **0.768** | **0.881** | **0.584** | **0.809** | **0.684** | 0.228 |

**EVOLMPNN outperforms all baselines on 8 of 9 splits.** Table 2 summarises performance comparison on `AAV`, `GB1` and `Fluorescence` datasets. EVOLMPNN achieves new state-of-the-art performance on most splits of three datasets, with up to $6.7\%$ improvements to baseline methods. This result vindicates the effectiveness of our proposed design to capture evolution information for homologous protein property prediction. Notably, GNN-based methods that utilise manually constructed graph structure do not enter top-2 on 9 splits and two Transformer structure models, *i.e.*, GraphTransformer and NodeFormer, often outperform such methods. It can be understood since homology graph construction is a challenging biomedical task (Pearson, 2013), the simple $K$-NN graph construction is not an effective solution.

**Large-scale PLM models are dominated by simple models.** Surprisingly, we find that smaller models, such as CNN and ResNet, can outpefoerm large ESM variaants pre-trained on million- and billion-scale sequences. For instance, ESM-1v has about 650 million parameters and is pre-trained on around 138 million UniRef90 sequences (Meier et al., 2021). Yet, CNN outperforms ESM-1v on three splits of `Fluorescence` dataset. This indicates the necessity of designs targeting specifically the crucial homologous protein engineering task.

**Our proposed extension models outperform all baselines on `GB1` dataset.** We performed additional experiments on `GB1` datasets to investigate the performance of two extended models, *i.e.*, EVOLGNN and EVOLFORMER and study the influence of different residue embedding initialisation methods. The results summarised in Table 3 evince that EVOLMPNN outperforms the other two variants in three splits, and all our proposed models outperform the best baseline. This result confirms the effectiveness of encoding evolution information for homologous protein property prediction. Besides, the models adopting the PLM encoding $\Phi^{\mathrm{PLM}}$ achieve better performance than

Table 3: Results on `GB1` datasets (metric: Spearman's $\rho$) of our proposed methods, with different residue embeddings. Top-2 performances of each split marked as **bold** and underline.

| Model | Split | | |
|---|---|---|---|
| | 2-vs-R. | 3-vs-R. | L.-vs-H. |
| Best Baseline | 0.757 | 0.878 | 0.548 |
| EVOLMPNN ($\Phi^{OH}$) | 0.766 | 0.877 | 0.553 |
| EVOLGNN ($\Phi^{OH}$) | 0.764 | 0.866 | 0.536 |
| EVOLFORMER ($\Phi^{OH}$) | 0.764 | 0.868 | 0.537 |
| EVOLMPNN ($\Phi^{PLM}$) | **0.768** | **0.881** | **0.584** |
| EVOLGNN ($\Phi^{PLM}$) | 0.767 | 0.879 | 0.581 |
| EVOLFORMER ($\Phi^{PLM}$) | 0.766 | 0.879 | 0.575 |

those using the one-hot encoding $\Phi^{OH}$. From this experiment, we conclude that residue information provided by PLM helps to capture protein's evolution information.

## 5.2 ANALYSIS OF PERFORMANCE

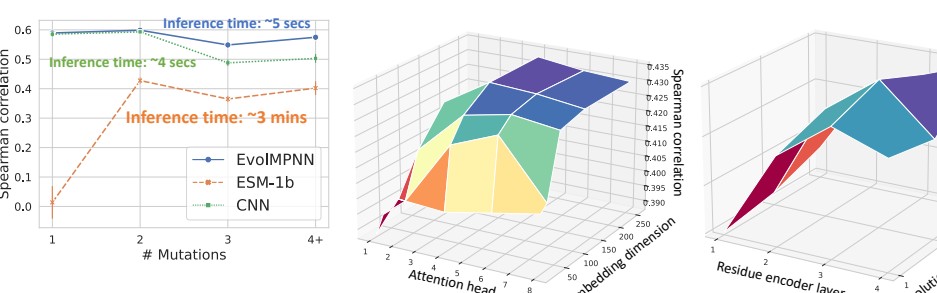

Figure 3: Performance on protein groups of different numbers of mutations, with the LOW-VS-HIGH split and avg. epoch inference time on `GB1` dataset.

(a) #Attention Heads & #Embedding Dimension   (b) Residue & Evolution Encoder #Layers

Figure 4: EVOLMPNN performance on `AAV`'s 2-VS-REST (a) and LOW-VS-HIGH (b) splits, with different hyper-parameters.

**The performance of EVOLMPNN comes from its superior predictions on high mutational mutants.** For the LOW-VS-HIGH split of `GB1` dataset, we group the test proteins into 4 groups depending on their number of mutations. Next, we compute three models, including EVOLMPNN, ESM-1b (fine-tuned PLM model) and CNN (best baseline), prediction performances on each protein group and present the results in Figure 3. EVOLMPNN outperforms two baselines in all 4 protein groups. Notably, by demonstrating EVOLMPNN's clear edge in groups of no less than 3 mutations, we confirm the generalisation effectiveness from low mutational mutants to high mutational mutants. **As per inference time**, EVOLMPNN and CNN require similar inference time ($\approx$ 5 secs), $36\times$ faster than ESM-1b ($\approx$ 3 mins).

**Influence of hyper-parameter settings on EVOLMPNN.** We present in Figure 4 a group of experiments to study the influence of some hyper-parameters on EVOLMPNN, including the number of attention heads, embedding dimension and the number of layers of residue encoder and evolution encoder. EVOLMPNN exhibits stable performance on different hyper-parameter settings.

## 6 CONCLUSION AND FUTURE WORK

We propose Evolution-aware Message Passing Neural Network (EVOLMPNN), that integrates both protein sequence information and evolution information by means of residues to predict the mutational effect on homologous proteins. Empirical and theoretical studies show that EVOLMPNN and its extended variants (EVOLGNN and EVOLFORMER) achieve outstanding performance on several benchmark datasets while retaining reasonable computation complexity. In future work, we intend to incorporate 3D protein structure information towards general-purpose homologous protein models.

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

## A    ALGORITHM

---

**Algorithm 1:** The framework of EVOLMPNN

---

**Input:** Protein set $\mathcal{M} = \{\mathcal{P}_i\}_{i=1,2,\ldots,M}$ and each protein sequence $\mathcal{P}_i$ contains a residue set $\{r_j\}_{j=1,2,\ldots,N}$; Message computation function $\mathrm{F}_{\mathrm{MESSAGE}}$ that outputs an $d$ dimensional message; COMBINE($\cdot$) and CONCAT($\cdot$) operators.

**Output:** Protein embeddings $\{\mathbf{Z}_i\}_{i=1,2,\ldots,M}$

1   $\mathbf{H}_i \leftarrow \mathrm{PLMENCODER}(\mathcal{P}_i)$

2   $\mathbf{X}_i \leftarrow \Phi^{\mathrm{OH}}(\{r_j\}_{j=1,2,\ldots,N}) \,/\, \Phi^{\mathrm{PLM}}(\{r_j\}_{j=1,2,\ldots,N})$

3   $\hat{\mathbf{X}}_i \leftarrow \mathbf{X}_i \odot \Phi^{\mathrm{POS}}$

4   $\mathbf{R}_i^{(0)} \leftarrow \hat{\mathbf{X}}_i$

5   **for** $\ell = 1, 2, \ldots, L_r$ **do**

6      **for** $i = 1, 2, \ldots, N$ **do**

7         $\mathbf{R}_i^{(\ell)} \leftarrow \mathrm{NODEFORMER}(\mathbf{R}_i^{(\ell-1)})$

8      **end**

9   **end**

10   $\mathbf{R}_i \leftarrow \mathbf{R}_i^{(L_r)}$

11   $\mathbf{H}_i^{(0)} \leftarrow \mathbf{H}_i$

12   **for** $\ell = 1, 2, \ldots, L_p$ **do**

13      $\{S_j\}_{j=1,2,\ldots,k} \sim \mathcal{M}$

14      **for** $i = 1, 2, \ldots, M$ **do**

15         **for** $j = 1, 2, \ldots, k$ **do**

16            $\mathbf{d}_{ij} = \mathrm{COMBINE}(\mathbf{R}_i - \mathbf{R}_{S_j})$

17         **end**

18         $\hat{\mathbf{H}}_i^{(\ell)} = \mathrm{COMBINE}(\{\mathrm{F}_{\mathrm{MESSAGE}}(i, j, \mathbf{H}_j^{(\ell)}, \mathbf{d}_{ij})\}_{j=1,2,\ldots,k})$

           $\mathbf{H}_i^{(\ell+1)} = \mathrm{CONCAT}(\mathbf{H}_i^{(\ell)}, \hat{\mathbf{H}}_i^{(\ell)})\mathbf{W}^\ell$

19      **end**

20   **end**

21   $\mathbf{Z}_i^{\mathrm{P}} = \mathbf{H}_i^{(L_p)}$

22   $\mathbf{Z}_i^{\mathrm{R}} = \mathrm{MEAN\ POOL}(\mathbf{R}_i)$

23   $\mathbf{Z}_i = \mathrm{CONCAT}(\mathbf{Z}_i^{\mathrm{P}}, \mathbf{Z}_i^{\mathrm{R}})$

---

We summarise the process of Evolution-aware Message Passing Neural Network (EVOLMPNN) in Algorithm 1. Given a protein set $\mathcal{M} = \{\mathcal{P}_i\}_{i=1,2,\ldots,M}$ and each protein sequence $\mathcal{P}_i$ contains a residue set $\{r_j\}_{j=1,2,\ldots,N}$. For each protein $\mathcal{P}_i$, we first initialise protein sequence ($\mathbf{H}_i$) and residue embeddings ($\mathbf{X}_i$) (line 1-2). After, the residue embeddings are empowered with positional encoding ($\Phi_{\mathrm{POS}}$) to get $\hat{\mathbf{X}}_i$ (line 3). Such a design will help us to record the position of a mutation occurring in the protein sequence in the following steps. Then, we update residue embedding based on a contact map, which records the chemical reactions between residues after folding into a 3D structure (line 4-10). Furthermore, we aggregate evolution-aware embeddings by means of updated residue embeddings (line 11- line 17) and integrate them with protein sequence embeddings to empower them with evolution-aware semantics (line 18-21). Finally, we merge protein sequence and residue embeddings as the final protein embeddings, which contain comprehensive information and make predictions based on them (line 22- line 23).

## B    THEORETICAL ANALYSIS

Inspired by (You et al., 2019), we adopt Bourgain's Theorem (Bourgain, 1985) to guide the random anchor number ($k$) of the evolution encoding layer, such that the resulting embeddings are guaranteed to have low distortion. Specifically, distortion measures the faithfulness of the embeddings in preserving distances (in our case, is the differences between protein sequences on a homology network) when mapping from one metric space to another metric space, which can be defined as:

**Definition 1** (Distortion). *Given two metric space $(\mathcal{M}, F_{\text{DIST}})$ and $(\mathcal{Z}, F'_{\text{DIST}})$ and a function $f :$ $\mathcal{M} \to \mathcal{Z}$, $f$ is said to have distortion $\alpha$, if $\forall \mathcal{P}_i, \mathcal{P}_j \in \mathcal{M}$, $\frac{1}{\alpha} F_{\text{DIST}}(\mathcal{P}_i, \mathcal{P}_j) \leq F'_{\text{DIST}}(f(\mathcal{P}_i), f(P_j)) \leq$ $F_{\text{DIST}}(\mathcal{P}_i, \mathcal{P}_j)$.*

**Theorem 1** (Bourgain Theorem). *Given any finite metric space $(\mathcal{M}, F_{\text{DIST}})$, with $| \mathcal{M} |= M$, there exists an embedding of $(\mathcal{M}, F_{\text{DIST}})$ into $\mathbb{R}^k$ under any $l_p$ metric, where $k = O(\log^2 M)$, and the distortion of the embedding is $O(\log M)$.*

Theorem 1 states the Bourgain Theorem (Bourgain, 1985), which shows the existence of a low distortion embedding that maps from any metric space to the $l_p$ metric space.

**Theorem 2** (Constructive Proof of Bourgain Theorem). *For metric space $(\mathcal{M}, F_{\text{DIST}})$, given $k = \log^2 M$ random sets $\{S_j\}_{j=1,2,\ldots,\log^2 M} \subset \mathcal{M}$, $S_j$ is chosen by including each point in $\mathcal{M}$ independently with probability $\frac{1}{2^j}$. An embedding method for $\mathcal{P}_i \in \mathcal{M}$ is defined as:*

$$f(\mathcal{P}_i) = (\frac{F_{\text{DIST}}(\mathcal{P}_i, S_1)}{k}, \frac{F_{\text{DIST}}(\mathcal{P}_i, S_2)}{k}, \cdots, \frac{F_{\text{DIST}}(\mathcal{P}_i, S_{\log^2 M})}{k}), \tag{9}$$

*Then, $f$ is an embedding method that satisfies Theorem 1.*

**Anchor Selection.** EVOLMPNN can be viewed as a generalisation of the embedding method of Theorem 2 (Linial et al., 1995), where $F_{\text{DIST}}(\cdot)$ is generalised via message passing functions (Eq 3-Eq. 6). Therefore, Theorem 2 offers a theoretical guide that $O(\log^2 M)$ anchors are needed to guarantee low distortion embedding. Following this principle, EVOLMPNN choose $k = \log^2 M$ random anchors, denoted as $\{S_j\}_{j=1,2,\ldots,\log^2 M}$, and we sample each protein in $\mathcal{M}$ independently with probability $\frac{1}{2^j}$.

## C  DATASETS

Adeno-associated virus (AAV) capsid proteins are responsible for helping the virus carrying viral DNA into a target cell (Vandenberghe et al., 2009); there is great interest in engineering versions of these proteins for gene therapy (Bryant et al., 2021; Büning et al., 2015; Barnes et al., 2019). (Bryant et al., 2021) produces mutants on a 28 amino-acid window from position 561 to 588 of VP-1 and measures the fitness of resulting variants with between 1 and 39 mutations. We adopt three splits from the benchmark (Dallago et al., 2021), including 2-VS-REST, 7-VS-REST and LOW-VS-HIGH.

GB1 is the binding domain of protein G, an immunoglobulin binding protein found in Streptococcal bacteria (Sauer-Eriksson et al., 1995; Sjöbring et al., 1991). Wu et al. (2016) measure the fitness of generated mutations. We adopt three splits from the benchmark (Dallago et al., 2021), including 2-VS-REST, 3-VS-REST and LOW-VS-HIGH.

The green fluorescent protein is an important marker protein, enabling scientists to see the presence of the particular protein in an organic structure by its green fluorescence (Tsien, 1998). Sarkisyan et al. (2016) assess the fitness of green fluorescent protein mutants. We adopt one available split, 3-VS-REST, from the benchmark (Xu et al., 2022). Besides, in order to evaluate the models' effectiveness, we add two splits, 2-VS-REST and LOW-VS-HIGH, in this paper.

## D  BASELINES

We present details about our baseline and proposed models in Table 4.

Table 4: Description and implementation of baseline methods.

| Method | Descriotion | Encoder | Pooling | Output layer |
|---|---|---|---|---|
| Levenshtein | Levenshtein distance to wild -type. | - | - | - |
| BLOSUM62 | BLOSUM62-score relative to wild-type. | - | - | - |
| DDE | ipeptide Deviation from Expected Mean | 2-layer MLP | - | - |
| Moran | Moran correlation | 2-layer MLP | - | - |
| LSTM | Simple LSTM model | 3-layer LSTM | Weighted sum pool | 2-layer MLP |
| Transformer | Simple Transformer model | 4-layer Transformer, 4 attention heads | - | 2-layer MLP |
| CNN | Simple convolutional model | 2-layer CNN | Max pool | 2-layer MLP |
| ResNet | Classic framework of skip connections and residual blocks | 8-layer ResNet | Attentive weighted sum | 2-layer MLP |
| ProtBert | 750M param transformer pre-trained on Uniref 50 | 30-layer BERT 16 attention heads | Linear pool | 2-layer MLP |
| ESM-1b | 650M param transformer pre-trained on Uniref 50 | - | Mean pool | 2-layer MLP |
| ESM-1v | 650M param transformer pre-trained on Uniref 90 | - | Mean pool | 2-layer MLP |
| ESM-2 | 3B param transformer pre-trained on Uniref 50 | - | Mean pool | 2-layer MLP |
| GCN | Graph convolutional network | 2/3-layer GCN encoder | - | 2-layer MLP |
| GAT | Graph attention network | 2/3-layer GAT encoder | - | 2-layer MLP |
| GraphTransf. | Transformer model designed for graphs | 2/3-layer GraphTransf. encoder | - | 1-layer MLP |
| Nodeformer | Efficient Transformer variant design for graphs | 2/3/4-layer Nodeformer encoder | - | 1-layer MLP |
| EVOLMPNN | - | 2/3-layer Residue encoder and Evolution encoder | - | 1-layer MLP |
| EVOLGNN | - | 2/3-layer Residue encoder and Evolution encoder | - | 1-layer MLP |
| EVOLFORMER | - | 2/3-layer Residue encoder and Evolution encoder | - | 1-layer MLP |

