# OpenReview forum: "EvolMPNN: Predicting Mutational Effect on Homologous Proteins by Evolution Encoding"
_ICLR.cc/2024/Conference — Submitted to ICLR 2024_

### Official Review · Reviewer_fHbu · 2023-10-17

**Soundness:** 3 good
**Presentation:** 3 good
**Contribution:** 2 fair
**Rating:** 3
**Confidence:** 4

**Summary:**

This paper proposes an EVOLMPNN, Evolution-aware Message Passing Neural Network, to learn evolution-aware protein embeddings. Specifically, it overcomes the drawback of existing methods that neglect subtle mutations and fail to capture their effect on the protein properties. The author claims a 6.4% promotion and a faster inference speed over the state-of-the-art method. Admittedly, the motivation of evolution information-based message passing is charming, but many details of this approach require further polished. I hope to see a better version of this algorithm and believe the present one cannot reach the high standard of ICLR.

**Strengths:**

(1) The evolution encoding is new and promising and presents a novel way to model relationships between different protein sequences.

(2) The figures are clearly drawn and helpful to understand the core idea of EvolMPNN.

(3) The author is very honest in posting all results of different settings even if EvolGNN and EvolFormer seem redundant.

(4) The code is available and made anonymous.

**Weaknesses:**

(1) Section 4.1  explains how to initialize the residue-level and protein-level embeddings. However, it is truly a combination of different PLMs and the logic is messy. It is said that a frozen PLM is used to extract protein sequence embedding, while another PLM (or the same PLM, I am not sure) is leveraged to attain residue embedding. After that, a position embedding is computed. But if the author has carefully read about the details of those PLMs, which almost are Transformer-based, they must know that PLMs have already introduced positional encoding. Why does the author depend on pretrained PLMs but initialize the embedding repeatedly using exactly the same methodology of PLMs? In Section 4.2, the author utilizes NodeFormer to further update the residue embedding, which I still cannot find any motivation. As proved in the original paper, NodeFormer is good at solving node classification problems in common graph datasets such as Cora, Citeseer, Deezer, and Actor. It is acceptable to see a protein sequence as a fully-connected graph and use NodeFormer to perform feature encoding. But the operation is completely the same as what PLMs have done. To summarize, Section 4.1 and 4.2 hope to replace PLMs with a different Transformer-based architecture to attain better embedding, but lacks novelty and innovations. Can the author provide an ablation study of NodeFormer and those embedding initialization mechanisms? Besides, can the author just directly tune the PLMs instead of stacking a number of extra Transformer layers?


(2) Though the author spends several pages describing the algorithm of EvolGNN, the key contribution is the evolution encoding part (Section 4.3). I like the idea of incorporating protein evolution information into the message-passing scheme. However, the selection of anchor is pretty rough and naive. As briefed by the author, homologous proteins typically have similarities in their amino-acid sequences, structures, and functions due to their shared ancestry. It is important to distinguish these subtle yet crucial differences for accurately predicting the homologous protein property. Therefore, it is a priority for us to pick up homologous proteins or at least similar protein sequences and combine their protein embeddings. However, the author uses a random selection mechanism to choose anchors, which violates the motivation of evolution encoding. And this is also why EvolGNN and EvolFormer fail (please see the third point below).


(3) EvolGNN and EvolFormer are two extensions of the foundation model EvolMPNN. However, Table 3 shows that EvolMPNN is better than EvolGNN and EvolFormer. This is really confusing and contradictory. Remarkably, The main difference between EvolMPNN and its two variants (i.e., EvolGNN and EvolFormer) is the incorporation of the evolutionary information (d_ij and R_i). However adding these two kinds of knowledge does not bring benefits over the basic EvolGNN. In my opinion, this does not confirm that encoding evolution information is useless. Instead, it illustrates that d_ij and R_i do not contain strong evolution information. Recall that the author randomly samples each protein from the entire dataset M independently with the same probability. The anchor and the target protein can be completely different and have little connection with each other. In other words, their edit distance is very large and cannot be regarded as wide-type and mutant sequences, respectively. Therefore, the addition of d_ij and R_i is useless.


(4) The results cannot convincingly support the superiority of the proposed method. First, GNN-based methods such as Graph Transformer and NodeFormer have already achieved competitive performance. The improvements over baselines are very limited. Second, the authors omit fine-tuning advanced PLMs (ESM-1b, ESM-2) due to some incomprehensible reason (48 hours of training time) and make the comparison unfair. I understand that the author faces a dilemma of limited computational resources, but this cannot be the excuse for not thoroughly running the experiments, particularly when PLMs are important in this problem.

**Questions:**

(1) In Table 2, some results (e.g., ProtBert and ESM-1b) are missing because the author states that they cannot complete the training process within 48 hours on their devices. However, I suppose this reason is not acceptable since I have never heard of 48 hours being a standard training time cutoff. I would encourage the authors to finish the training of those PLMs and compare EvolMPNN with them fairly.

Besides, it can be discovered that fine-tuned ESM-1b is much better than fixed ESM-1b (with 650M parameters). Therefore, it would be necessary to investigate the upper limit of PLMs by fine-tuning ESM-2 (with 3B or even 15B) on this task. Notably, fine-tuned ESM-1b achieves outstanding results in all three datasets (i.e., Fluorescence, AAV, and GB1). It is highly possible that fine-tuned ESM-2 can outperform EvolMPNN. I would highly recommend the authors use more powerful computational resources such as A100 with 80GB instead of 3090 GPU with only 24GB to implement fine-tuning large-scale PLMs. Last but not least, it would be interesting to test some latest PLMs such as Ankh, which outpasses ESM-2 in a range of downstream problems.

[A] Ankh: Optimized Protein Language Model Unlocks General-Purpose Modelling

(2) The one-hot embedding and PLM embedding in the residue embedding block are separate from each other. Meanwhile, in Table 3, one-hot embedding is completely inferior to PLM embeddings, so what is the meaning of introducing this one-hot embedding? This comparison is not appealing and I would suggest the authors delete the one-hot embedding part or move it to the Appendix.

(3) In Figure 3, the author claims that EvolMPNN is much faster than PLMs (namely, ESM-1b) during the inference time. I am really confused about this point.   As mentioned in Section 4.1, EvolMPNN relies on PLMs to extract protein sequence embedding as well as residue embedding. What sort of PLMs has the author used for this feature extraction? If the author uses the same ESM-1b, EvolMPNN should definitely require a longer inference time than PLMs.

(4) In Figure 4, the author points out that EvolMPNN exhibits stable performance on different hyper-parameter settings. However, it can be found that in some bad hyperparameter settings, EvolMPNN is not state-of-the-art, which outlines the importance of hyperparameter search. Therefore, I wonder whether the author has conducted a hyperparameter search when reproducing the baselines such as Graph Transformer and NodeFormer.

Moreover, I suppose there is some mistake in Figure 4's caption: subfigure (a) reports the performance of AAV's Low-vs-High while (b) reports AAV's 2-vs-Rest.

---

### Official Review · Reviewer_LeqQ · 2023-11-01

**Soundness:** 3 good
**Presentation:** 2 fair
**Contribution:** 2 fair
**Rating:** 3
**Confidence:** 4

**Summary:**

This work proposed an evolution-aware encoding method for mutational effect prediction. It defines a relational graph for homologous proteins and leverages MPNN to analyze the relationship among these proteins for properties prediction.

**Strengths:**

This work innovatively designs a relational network on the protein level with relevant proteins in its homologous family. The motivation is clearly introduced, and the methodology is different from other existing solutions for fitness prediction.

**Weaknesses:**

1. Notations are defined unclearly. I have to go back and forth to understand the method.
2. Many terminologies are different from their conventional meanings in literature, which brings unnecessary understanding barriers to the readers. For instance, homologous proteins usually include proteins with a relatively high sequence identity (i.e., 30%), which are not derived from protein engineering. Also, 'wild-type' proteins usually refer to naturally existing proteins, instead of ancestor proteins.
3. Protein language models do not only count MSA-based methods, and they certainly are not the only type of language models that perform mutational effect predictions, e.g., ESM-1v and Tranception.
4. Experiments are incomplete. A very limited category of proteins and deep learning methods were examined. For instance, the 87 proteins in ProteinGym should have been explored. Also latest fitness predicting methods (such as those reported at https://github.com/OATML-Markslab/ProteinGym)

**Questions:**

1. Page 3: "Jumper et al. (2021) explore MSA to capture the mutational effect." What did AF2 do to “capture the mutational effect”?
2. Section 4.1: "Residue Embedding". What is r? Also why with the same PLMEncoder() and the same output dimension (d), eq(1) embeds macro-level info, but here \Phi^{PLM} encodes micro-level info?
3. Page 5: Why does ‘the most parsimonious tree’ have to be found for defining a protein family graph with evolutionary patterns?
4. Page 5: What is the empirical evidence for "using MEAN Pool" and "simple element-wise product"?

---

### Official Review · Reviewer_ojZm · 2023-11-04

**Soundness:** 3 good
**Presentation:** 3 good
**Contribution:** 3 good
**Rating:** 6
**Confidence:** 3

**Summary:**

This submission proposes an evolution aware message passing network to predict the mutational effect on a protein by learning a suitable embedding.  Instead of using an exisitng phylogenetic graph/tree between proteins or building one, this submission proposes to learn this evolutionary information on the fly using anchor proteins and then aggregates this information with a message passing neural network. The resulting network, EvolMPNN is then evaluated on several benchmark datasets where it outperforms many baselines in protein property prediction task.

**Strengths:**

- the submission provides a novel idea of studying mutational effect on a protein without using an existing phylogentic tree. the presentation of the methodology is clear and well written. Complexity analysis is also appreciated. (more to come)

- the idea is well executed and compiled. the problem is well motivated and the choice of baselines (EvolGNN, EvolFormer) is well designed and justified. the evaluation seem exhaustive and convincing. the submission also highlights various insights wrt baselines in the experimental section.

- the experimental section provides various insights not just for this method but also several PLM and other baselines. e.g. large scale PLM  such as ESM1B being dominated by simple models (CNN) is an interesting insight given how ubiquitous ESM models are in protein embedding literature.

**Weaknesses:**

- the abstract or even the experiment section nowhere mentions clearly what the protein embeddings are used for. e.g. the abstract describes the MPNN method to obtain protein embeddings and then switches to model performance and state of the art without mentioning what is being predicted. Research problem section in related work contains a paragraph on the same. however it could be introduced much earlier and some of related work description can be pushed back in the related work instead of introduction.

- Similarly section 5 Experimental Study describes dataset, baselines and implementation without mentioning the output that the method predicts on these datasets.

- the submission attributes the performance gain to its superior predictions on high mutational mutants but does not provide an intuituin or justification for it. (more to come)

**Questions:**

- I did not find the value of M (no of anchor proteins) chosen in algorithm. this is an important hyperparameter also in terms of complexity. perhaps, the submission should mention it also in the complexity analaysis to get a sense how far M is from N. if M~N, then MPNN becomes quadratic in complexity. Similarly i did not find the ablation wrt this parameter.

-  <mutants of   no more than \lambda no of mutations> does it mean the set contains a max. \lambda mutations from the wild type ? if so , what is the avg max no. of mutations in general in a set?

- ProteinMPNN is another prominent work  that is designed for protein engineering/design and considered state of the art in this domain. However, i did not find any mention or comparison with it. Is it not applicable to the benchmarks considered here? please clarify.

---

### Meta-Review · Area_Chair_mX5t · 2023-12-21

**Metareview:**

This work presents EvolMPNN, a neural network designed for predicting protein mutation effect through the generation of evolution-aware protein embeddings. It selects anchor proteins, analyzes evolutionary data through residues, and then understand mutations' effects relative to these anchors. This method has the potential to enhance the understanding of mutations in protein properties. The execution and compilation of the code is well documented, and the proposed approach was benchmarked with reasonable baseline models (EvolGNN and EvolFormer). The evaluation process appears to showcase the advantage of the method. It is noteworthy that the authors also should be commended for presenting very thoughtful analysis and discussion of existing pre-trained protein language models, including the performance of ESM-based models could still have significant gap in terms of how they would be generally applicable for function prediction, a very challenging task as the authors noted. The major weakness of this work comes from (1) the confusion from inaccurate usage of terminology and concepts in the submission; (2) unclear description of the process and details related to the method and tests performed, as noted by several reviewers; (3) the limited set of validation that could convincingly showcase the novelty and broad applicability of the proposed EvolMPNN.

**Justification For Why Not Higher Score:**

The authors did not engage in discussion or revision with the reviewers, so at its current form it would not be suitable for acceptance. A revised version of this work would be more competitive as a formal submission in the future.

**Justification For Why Not Lower Score:**

NA

---

### Decision · Program_Chairs · 2024-01-16

Reject